# Cadmium Accumulation in Cacao Plants (*Theobroma cacao* L.) under Drought Stress

Antonio Ortiz-Álvarez [1,*], Stanislav Magnitskiy [1], Elías Alexander Silva-Arero [2], Caren Rodríguez-Medina [2], Xavier Argout [3,4] and Ángela María Castaño-Marín [2]

[1] Facultad de Ciencias Agrarias, Universidad Nacional de Colombia, Sede Bogotá, Bogotá 111321, Colombia; svmagnitskiy@unal.edu.co

[2] Corporación Colombiana de Investigación Agropecuaria—Agrosavia Km 14 Vía Mosquera, Bogotá 250047, Colombia; esilva@agrosavia.co (E.A.S.-A.); cdrodriguez@agrosavia.co (C.R.-M.); amcastano@agrosavia.co (Á.M.C.-M.)

[3] Centre de Coopération International en Recherche Agronomique Pour le Développment—CIRAD, UMR AGAP Institut, F-34398 Montpellier, France; xavier.argout@cirad.fr

[4] UMR AGAP Institut, Univ Montpellier, CIRAD, INRAE, Institut Agro, F-34398 Montpellier, France

* Correspondence: anortizal@unal.edu.co

**Abstract:** The objective of this study was to determine Cd accumulation under water-deficit conditions by young cacao plants. The study was conducted under greenhouse conditions. Two full-sib families (IMC67 × PA121 and SCA6 × PA121), obtained through controlled crosses, and an open pollinated half-sib seedling family of IMC67, widely used as rootstock in Colombia, were employed. Plants were grown in Cd-contaminated soil (0.356 mg kg$^{-1}$) without external sources of the metal. They were subjected to water deficit by suspending irrigation for consecutive periods of 19 and 27 days (D19 and D27), followed by rehydration. Water stress reduced leaf water potential ($\Psi_{leaf}$) with values from −1.51 to −2.09 MPa, with full-sib family SCA6 × PA121 being the most tolerant to water deficit. Cd concentration was influenced by biomass reduction (observed in IMC67 × PA121 and SCA6 × PA121) and transpiration rate (evident in IMC67) caused by water deficit. Full-sib progenies IMC67 × PA121 and SCA6 × PA121 accumulated more Cd in the plants than open pollinated IMC67, with higher Cd accumulation in leaves. The translocation factor (TF) revealed that the aboveground organs of the progenies were enriched with Cd (TF > 4). Water deficit increased Cd translocation from roots to leaves in IMC67 × PA121 and IMC 67, while there were no significant changes in SCA6 × PA121. Full-sib family SCA6 × PA121 stood out as the most promising progenies due to their water-stress tolerance and Cd accumulation stability. This study introduces a discussion about the influence of water stress on Cd accumulation in *Theobroma cacao*.

**Keywords:** *Theobroma cacao*; water stress; genotype; progeny; tolerance; Cd uptake; Cd translocation





## 1. Introduction

Climate change is one of the challenges currently facing agriculture as it results in extreme events, such as droughts and floods. The increase in the frequency of dry seasons may cause a decrease in crop production in the coming years, leading to a threat to food security [1]. In a study conducted in Bahia, Brazil, it was demonstrated that the strong drought in 2015–2016 caused the loss of 15% of cacao trees, resulting in a 89% decrease in yield [1]. A study using physiological modeling analyzed climatic information from ten cacao-producing countries and showed a reduction in production of up to 50% in regions characterized by dry seasons [2]. Several studies have shown that water deficit causes a negative impact on the growth and photosynthesis of cacao plants [1–4].

Regions in South and Central America host cocoa-growing areas, where the highest levels of Cd in cocoa beans have been recorded compared to other parts of the world, with values ranging between 0.6 and 0.8 mg kg$^{-1}$ [5]. These concentrations can surpass

the maximum allowable limits set by the European Union, which establishes a maximum tolerated limit of 0.8 mg kg$^{-1}$ for final products with cacao solids equal to or exceeding 50% [5–7]. This situation raises significant concerns for the quality and safety of cocoa-derived products.

Various studies have reported that tolerance to Cd accumulation is based on genotypic differences [8,9].While some previous studies have investigated genetic variation in Cd accumulation and distribution in cocoa. These studies have been limited in terms of the number of genotypes evaluated, the environmental conditions in which they were conducted, and the lack of information on Cd bioaccumulation in cacao [10]. Additionally, most studies on Cd absorption and accumulation in young cacao plants are carried out using soils or substrates artificially enriched with soluble sources of Cd, such as cadmium chloride ($CdCl_2$) [8,11] and cadmium nitrate (Cd ($NO_3$)$_2$) [12]. Likewise, studies conducted in Cd-contaminated soils without the addition of specific sources of this heavy metal have been conducted in plantations with an age exceeding ten years [10,13–15].

To date, no studies have addressed the interaction between water deficit and Cd accumulation in cacao. However, in species such as castor bean, Indian mustard, and tomato, among others, it has been observed that Cd absorption and accumulation in plant tissues tend to decrease based on plant transpiration under water-deficit conditions [16–18]. On the other hand, in wheat, soybean, and peanut, it has been reported that Cd concentration tends to increase due to reduced biomass accumulation under water-stress conditions [19–23].

The absorption and accumulation of Cd under water-deficit conditions appear to largely depend on genotype and its tolerance to this condition. According to Bashir et al. (2019), tolerant soybean cultivars are less affected in terms of root length, area, and diameter compared to susceptible ones, allowing them to absorb a greater amount of Cd from the soil [21]. In the case of tomato species, Ünyayar et al. (2005) found that Cd accumulated to a greater extent in the roots of the drought-sensitive species than in the drought-tolerant one, resulting in differences in Cd translocation to the aboveground part of the plants [18]. The authors concluded that a clear relationship exists between Cd absorption and drought tolerance [18]. Conversely, for castor bean, Shi et al. (2015) reported that Cd translocation from the roots to aboveground organs was not affected by water stress [17].

The preceding analysis reveals significant differences between species and genotypes in Cd accumulation under water-deficit conditions. Therefore, it is considered essential to study this behavior in cacao plants, given that water deficit and the presence of Cd in the soil are becoming increasingly common in cacao-growing areas of Colombia and the world due to climate change and soil pollution by heavy metals like Cd [24–26].

In this regard, the objective of this study was to evaluate the influence and interaction of genotype and soil water content on Cd accumulation and distribution in cacao plants. Unlike other studies that use substrates or soils artificially enriched with soluble sources of Cd, such as Cd ($NO_3$)$_2$ and $CdCl_2$ [8,11,12], this research was conducted under conditions closer to field reality, in which plants grew without the addition of external Cd sources.

## 2. Materials and Methods

### 2.1. Study Site

The experiment was conducted under greenhouse conditions at the Nataima Research Center of AGROSAVIA, located in a tropical dry forest area at 4°11′31″ N and 74°57′41″ W, with an altitude of 418 m above sea level in the municipality of El Espinal, Tolima, Colombia.

### 2.2. Plant Materials

Plant material was obtained at the Palmira Research Center of AGROSAVIA through controlled crosses between accessions from the Colombian Cacao Germplasm Bank, IMC67 × PA121 and SCA6 × PA121. An open pollinated (OP) half-sib seedling family of IMC67, widely used as rootstock in Colombia, was also considered.



The obtained seeds were germinated and maintained in an inert medium (river sand) for 60 days at the Nataima Research Center—AGROSAVIA. Subsequently, two-month-old plants with 6 to 8 true leaves were transplanted into plastic bags containing soil, where they remained until the end of the experiment. The soil used had an average soluble Cd content of 0.356 mg kg$^{-1}$, determined by inductively coupled plasma optical emission spectrometry (ICP-OES).

### 2.3. Planting and Growth

Under greenhouse conditions, transplanting was conducted using black plastic bags (60 cm in height × 30 cm in diameter) filled with 40 kg of loam soil. The soil had an average pH of 5.29 and an organic carbon content of 5.1%. Prior to transplanting, the soil was homogenized through a 12 mm sieve. Based on soil analysis results, each plant, at four months of age, was fertilized with 5.7 g of Calcinit B (15.5% N; 26.0% Ca; 0.1% B), 8.6 g of DAP (18.0% N; 46% P), 5.7 g of ManuKiesek (3% K; 24% Mg; 18% S), and 9 mL of Transfer ionic (an organic complex of carboxylic acids, gluconates, and ascorbic acid). This aimed to adjust ionic relationships and base saturation in the soil according to cacao nutritional requirements [27]. From transplantation until the initiation of water-stress treatments, all plants were hydrated to field capacity using tap water. Irrigation was adjusted based on soil moisture, monitored through CS616 volumetric moisture sensors (Campbell Scientific$^{®}$, Logan, UT, USA). Each plant received between 250 and 300 mL of water with an average frequency of every two days, depending on detected soil moisture conditions.

Water-deficit treatments were applied when the plants reached six months of age [28]. The experimental design consisted of a randomized complete block with three replications, following a factorial arrangement. The first factor comprised the progenies IMC67 × PA121, SCA6 × PA121 and OP IMC67, while the second factor included two water status states: plants subjected to water-deficit stress (DS) and well-watered plants (WW). For the WW treatment, plants were optimally irrigated to maintain a soil volumetric water content (VWC) between 38% and 40% during the evaluation period. The soil field capacity was determined through the soil moisture retention curve of the experiment. VWC was monitored using calibrated CS616 volumetric moisture sensors (Campbell Scientific$^{®}$, USA) placed at a depth of 25 cm in the soil, corresponding to the region of secondary and absorptive roots responsible for water and nutrient uptake by cacao plants [29].

### 2.4. Drought Stress Treatments

Predawn leaf water potential ($\Psi_{leaf}$) was maintained between $-0.2$ and $-0.7$ MPa, consistent with reported values for optimally hydrated cacao plants (Dos Santos et al. [28]). In the DS treatment, irrigation was withheld for consecutive periods of 19 (D19) and 27 (D27) days until $\Psi_{leaf}$ reached values ranging from $-1.4 \pm -0.4$ to $-2.05 \pm 0.25$ MPa, respectively. This deliberate action subjected the plants to moderate and severe levels of water stress [28,30,31], during which the VWC was maintained at approximately 30% and 25%, respectively. Following the D19 and D27 stress phases, all plants were rehydrated over a three-day period until reaching field capacity values of 38–40%, thus allowing for recovery from the stress phase (rehydration phase—DRH). Evaluations of recovery were conducted on the third day after rehydration (DRH).

All parameters were assessed at three distinct time points: during the 19th and 27th days of water stress (D19 and D27) and at the conclusion of the rehydration phase (DRH). However, biomass and Cd content in the plants were exclusively determined on D19 and D27. For these measurements, leaf tissue samples were frozen in liquid nitrogen and subsequently stored at $-80$ °C. These samples were later utilized to determine the physiological and biochemical variables of the cacao plants.

### 2.5. Environmental Conditions during the Experiment

During the course of the experiment, measurements of key environmental variables were conducted, including air temperature (°C), relative humidity (% RH), and solar

radiation (W m$^{-2}$). For this purpose, two ATMOS 14 sensors from Meter Groups (Pullman, WA, USA) were used, adapted to an EM50 Data Logger, and placed inside the greenhouse. These sensors recorded temperature and relative humidity on a daily basis, with a sampling frequency of 30 min. Additionally, to measure solar radiation, a Vantage Pro2 weather station (Davis Instruments, Newberry, FL, USA) was employed, situated at a distance of 200 m from the greenhouse. The data collected throughout the entire experiment are presented in Figure 1.

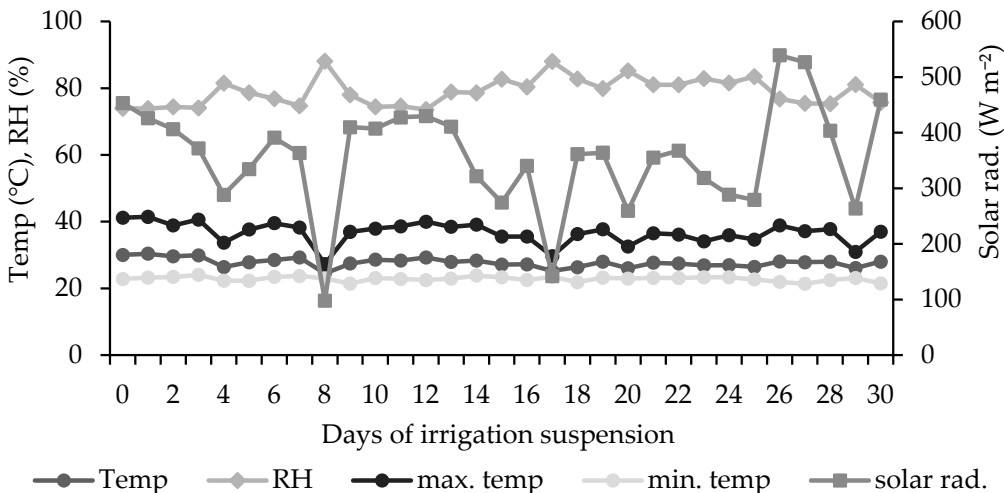

**Figure 1.** Daily average temperature inside the greenhouse (Temp), relative humidity (RH), maximum temperature (max. temp), minimum temperature (min. temp) inside the greenhouse and solar radiation (solar rad.) conditions to which cacao plants were exposed throughout the duration of the experiment.

### 2.6. Leaf Water Potential ($\Psi_{leaf}$)

The leaf water potential ($\Psi_{leaf}$) was determined between 2:00 a.m. and 4:00 a.m. by sampling the third mature leaf from the top to the bottom of the plant [30]. Nine plants per treatment ($n = 9$) were selected, and measurements were taken using a Schölander pressure chamber (PMS Model 615, Fresno, CA, USA).

### 2.7. Gas Exchange

The parameters of photosynthetic rate (*A*), stomatal conductance (*gs*), and transpiration (*E*) were recorded from 9:00 a.m. to 12:00 p.m. on the fourth mature leaf from the top to the bottom of the plant [32]. Twelve plants per treatment ($n = 12$) were selected for measurements using a portable photosynthesis system, LI-6800XT (LI-COR Biosciences Inc., Lincoln, NE, USA). The measurements were taken at an ambient $CO_2$ concentration of 400 μmol m$^{-2}$ s$^{-1}$ [30] and a photosynthetic photon flux density of 600 μmol m$^{-2}$ s$^{-1}$.

### 2.8. Biomass Accumulation

Six plants per treatment ($n = 6$) were selected and divided into leaves + petioles, stems, and roots. These plant parts were placed in an oven at 60 °C for three days to determine their dry weight using an analytical balance. The root-to-shoot ratio (R/S) of the plant was calculated by dividing the dry weight of the roots by the dry weight of the leaves + stems.

### 2.9. Soil Cd Content

The soil used for plant growth in the experiment was extracted from an agricultural plot located at the Tibaitatá Research Center of AGROSAVIA. This soil was specifically chosen for its initial pseudototal Cd content (4.85 mg kg$^{-1}$) and its physicochemical properties suitable for cacao plant development. Given the soil's historical use over the decades, the

presence of Cd is considered anthropogenic in origin, attributed to agricultural activities, particularly the use of phosphoric fertilizers that enrich the soil with this metal.

To ensure uniformity, a total of 10.08 tons of soil were homogenized and used to fill black plastic bags, each with a capacity of 40 kg, measuring 65 cm in height and 28 cm in diameter. These bags were subsequently used for transplanting cacao seedlings with six to eight true leaves.

The Cd content in the soil was determined following the methodology described by Bravo et al. (2021) [24].

Based on the results of this determination, it can be affirmed that plants from all treatments, including well-watered plants, grew in soil with an average soluble Cd content of 0.356 mg kg$^{-1}$, a value considered high according to Dutta et al. (2020) [33].

### 2.10. Plant Cd Content

To determine the Cd content in the plant, organs were separated into leaves with petioles, stems, and roots from three plants per progeny (*n* = 3) on days 19 and 27 of stress. Subsequently, the samples were dried in an oven at 70 °C for one week until reaching a constant weight. The Cd content in roots, stems, and leaves was obtained using inductively coupled plasma optical emission spectroscopy (ICP-OES) [34]. The results of the plant Cd content were expressed in terms of concentration in mg Cd per kg of the organ.

To determine the Cd accumulation per plant organ, the Cd concentration was multiplied by the dry mass of each organ, expressed in mg of Cd per organ. The total Cd accumulation in the plant was obtained by summing the Cd accumulation in roots, stems, and leaves.

### 2.11. Translocation Factor (TF)

The Cd Translocation Factor (TF) index was determined using the equation by Mattina et al. (2003) (see Equation (1)), as employed by De Almeida et al. (2022) in 8-month-old cacao plants [11,35].

$$\text{TF} = \frac{\text{Cd concentration in shoots}}{\text{Cd concentration in roots}} \tag{1}$$

### 2.12. Statistical Analysis

A two-way ANOVA (progeny × stress treatment) was conducted to identify significant differences between treatments using the Tukey's honestly significant difference (HSD) test. In addition, the Cd content in the leaves was included in the Pearson correlation analysis along with other physiological parameters to determine the relationship between Cd accumulation and these variables.

## 3. Results

### 3.1. Leaf Water Potential ($\Psi_{leaf}$)

The leaf water potential ($\Psi_{leaf}$) of cacao plants was significantly different between well-watered (WW) and stressed (DS) treatments (Table 1). In well-watered (WW) plants, $\Psi_{leaf}$ ranged between −0.31 and −0.61 MPa, while in stressed (DS) plants, $\Psi_{leaf}$ ranged between −1.51 and −2.09 MPa. On day 19 (D19) of water stress, the progeny that exhibited the least reduction in $\Psi_{leaf}$ was SCA 6 × PA121 (−1.51 MPa), followed by IMC67 OP half-sib (−1.78 MPa), and IMC 67 × PA121 full-sib familiescA2 (−1.87 MPa).

The lowest $\Psi_{leaf}$ values were observed in DS plants on D27 (−2.04 ± 0.05 MPa) when VWC reached 24.6% (Table 1), with no significant differences among progenies. However, three days after rehydration (DRH), all progenies previously subjected to stress (DS) showed $\Psi_{leaf}$ values similar to the WW treatment.

**Table 1.** Leaf water potential $\Psi_{leaf}$ (MPa) of three cacao family progenies under water stress and rehydration.

| Progeny | | | $\Psi_{leaf}$ (MPa) | |
|---|---|---|---|---|
| | | Water Stress | DS | DRH |
| IMC67 × PA121 | D19 | WW | −0.52 ± −0.10 ef | −0.34 ± −0.04 FG |
| | | DS | −1.87 ± −0.11 bc | −0.33 ± −0.04 FG |
| | D27 | WW | −0.37 ± −0.12 fg | −0.42 ± −0.04 DE |
| | | DS | −2.06 ± −0.13 a | −0.52 ± −0.06 BC |
| SCA6 × PA121 | D19 | WW | −0.47 ± −0.10 fg | −0.29 ± −0.06 G |
| | | DS | −1.51 ± −0.06 de | −0.37 ± −0.05 EF |
| | D27 | WW | −0.31 ± −0.07 g | −0.61 ± −0.05 A |
| | | DS | −2.09 ± −0.08 a | −0.62 ± −0.07 A |
| IMC67 | D19 | WW | −0.63 ± −0.06 e | −0.33 ± −0.06 EF |
| | | DS | −1.78 ± −0.13 c | −0.47 ± −0.06 CD |
| | D27 | WW | −0.37 ± −0.10 fg | −0.57 ± −0.07 AB |
| | | DS | −1.98 ± −0.10 ab | −0.54 ± −0.05 ABC |

WW: Well-watered; DS: water-deficit stress; DRH: rehydration of plants subjected to D19 and D27. Lowercase letters indicate the significance of the treatments in the water-deficit stress phase (DS), while uppercase letters represent the significance of the treatments in the rehydration phase (DRH). Average values ($n = 9$). Significant differences ($p < 0.05$) using two-way ANOVA (progeny × stress). Different letters indicate significant differences according to Tukey's HSD test.

*3.2. Gas Exchange*

Water-deficit stress affected the ability of cacao plants to exchange gases. In the DS treatment, photosynthesis (*A*) values ranged from 5.92 to 9.08 μmol $CO_2$ $m^{-2}$ $s^{-1}$, and significant differences were observed among progenies and the days of irrigation suspension (D19 and D27) (Table 2).

**Table 2.** Gas exchange parameters in three cacao family progenies under water stress and rehydration.

| Progeny | Water Stress | | *A* (μmol $CO_2$ $m^{-2}$ $s^{-1}$) | *gs* (mol $H_2O$ $m^{-2}$ $s^{-1}$) | *E* (mmol $H_2O$ $m^{-2}$ $s^{-1}$) |
|---|---|---|---|---|---|
| IMC67 × PA121 | D19 | WW | 7.46 ± 0.61 c | 0.07 ± 0.02 ab | 0.00167 ± 0.00032 ab |
| | | DS | 3.47 ± 0.53 e | 0.03 ± 0.01 c | 0.00077 ± 0.00033 fg |
| | DRH | WW | 6.74 ± 0.7 B | 0.07 ± 0.02 ABC | 0.00153 ± 0.0005 ABC |
| | | DS | 6.72 ± 0.63 BC | 0.07 ± 0.01 ABC | 0.00151 ± 0.00034 ABC |
| | D27 | WW | 8.75 ± 0.41 ab | 0.08 ± 0.02 a | 0.00129 ± 0.00033 bcd |
| | | DS | 2.14 ± 0.57 gh | 0.02 ± 0.01 c | 0.00037 ± 0.00021 gh |
| | DRH | WW | 7.43 ± 0.8 AB | 0.07 ± 0.02 ABC | 0.00135 ± 0.00065 ABC |
| | | DS | 7.31 ± 0.79 AB | 0.07 ± 0.02 ABC | 0.00148 ± 0.00076 ABC |
| SCA6 × PA121 | D19 | WW | 8.21 ± 0.63 b | 0.07 ± 0.02 ab | 0.00171 ± 0.00037 a |
| | | DS | 3.88 ± 0.47 e | 0.03 ± 0.01 c | 0.00085 ± 0.00024 ef |
| | DRH | WW | 7.45 ± 0.63 AB | 0.08 ± 0.02 A | 0.00172 ± 0.00052 A |
| | | DS | 6.71 ± 0.68 BC | 0.05 ± 0.01 C | 0.00088 ± 0.00025 C |
| | D27 | WW | 9.08 ± 0.48 a | 0.09 ± 0.02 a | 0.00121 ± 0.0002 cde |
| | | DS | 2.65 ± 0.61 fg | 0.03 ± 0.02 c | 0.00058 ± 0.00028 fgh |
| | DRH | WW | 8.17 ± 0.83 BC | 0.08 ± 0.02 AB | 0.0018 ± 0.00078 AB |
| | | DS | 6.89 ± 0.75 B | 0.07 ± 0.01 ABC | 0.00122 ± 0.00051 ABC |
| IMC67 | D19 | WW | 5.92 ± 0.5 d | 0.06 ± 0.02 b | 0.00148 ± 0.00043 abc |
| | | DS | 3.29 ± 0.37 ef | 0.03 ± 0.01 c | 0.00089 ± 0.00029 def |
| | DRH | WW | 5.6 ± 0.68 D | 0.05 ± 0.02 BC | 0.00093 ± 0.00032 C |
| | | DS | 5.82 ± 0.65 DC | 0.05 ± 0.01 C | 0.00089 ± 0.00021 C |
| | D27 | WW | 6.33 ± 0.52 d | 0.07 ± 0.02 ab | 0.00122 ± 0.00042 cde |
| | | DS | 1.54 ± 0.47 h | 0.02 ± 0.01 c | 0.00031 ± 0.0002 h |
| | DRH | WW | 6.55 ± 0.69 BC | 0.06 ± 0.03 ABC | 0.00108 ± 0.00043 BC |
| | | DS | 6.69 ± 0.6 BC | 0.07 ± 0.02 ABC | 0.00128 ± 0.00035 ABC |

*A*, Photosynthetic rate; *gs*, stomatal conductance; *E*, transpiration; WW, well-watered; DS, water-stress phase; DRH, rehydration phase; D19, 19 days of water stress; D27, 27 days of water stress. Lowercase letters indicate the significance of the treatments in the water-deficit stress phase (DS), while uppercase letters represent the significance of the treatments in the rehydration phase (DRH). Average values ($n = 12$). Significant differences ($p < 0.05$) determined by two-way ANOVA (progeny × stress). Different letters indicate significant differences according to Tukey's HSD test.



Furthermore, in the stress treatment (DS), *A* decreased by 50.20% in plants subjected to 19 days without irrigation (D19) ($3.55 \pm 0.33$ µmol $CO_2$ m$^{-2}$ s$^{-1}$) and even by 74.00% in those subjected to 27 days ($2.11 \pm 0.57$ µmol $CO_2$ m$^{-2}$ s$^{-1}$). Significant differences were only found at the highest stress level (D27) between IMC 67 OP half-sib and SCA6 × PA121 full-sib progenies. The variable *gs* did not show differences *among* progenies after 19 and 27 days of stress. However, IMC67 OP half-sib progenies exhibited a significant 64.90% reduction when plants were left without irrigation for 27 days (D27), compared to those stressed for 19 days (D19). After rehydrating stressed plants (DRH), *A*, *gs*, and *E* values were restored, on average, to 94.2% of the values observed in well-watered (WW) plants (Table 2).

### 3.3. Biomass Accumulation

There was a lower biomass accumulation observed in cacao plants subjected to water-stress treatments (DS) compared to well-watered plants (WW) (Table 3). However, SCA6 × PA 121 full-sib progenies were a unique exception, as it accumulated only 6.82% less bio-mass than control plants (BR) when subjected to 19 days without irrigation (D19), without significant differences.

**Table 3.** Biomass accumulate (g) in plant organs of three family progenies under water stress.

| Progeny | Water Stress | | Biomass (g) | | | | |
|---|---|---|---|---|---|---|---|
| | | | Leaves | Stem | Roots | R/S | Total |
| IMC67 × PA121 | D19 | WW | 45.58 ± 2.83 abc | 25.78 ± 2.25 bc | 17.36 ± 1.68 a | 0.24 ± 0.03 a | 88.72 ± 1.42 ab |
| | | DS | 41.44 ± 2.89 cd | 22.58 ± 1.15 cd | 15.78 ± 2.27 abc | 0.25 ± 0.03 a | 79.81 ± 4.45 cd |
| | D27 | WW | 45.86 ± 3.93 abc | 30.08 ± 2.09 a | 16.13 ± 0.74 ab | 0.21 ± 0.02 abc | 92.07 ± 2.12 ab |
| | | DS | 42.02 ± 2.69 bcd | 23.96 ± 1.85 cd | 13.32 ± 2.27 bcd | 0.20 ± 0.04 abcd | 79.29 ± 3.99 cd |
| SCA6 × PA121 | D19 | WW | 48.38 ± 2.93 abc | 23.08 ± 1.56 cd | 13.35 ± 2.10 bcd | 0.19 ± 0.03 bcd | 84.82 ± 2.97 bc |
| | | DS | 42.28 ± 3.53 bcd | 20.74 ± 1.92 d | 12.44 ± 1.47 cd | 0.20 ± 0.03 abcd | 75.46 ± 3.26 cd |
| | D27 | WW | 50.35 ± 2.66 abc | 29.10 ± 2.22 ab | 13.12 ± 2.34 bcd | 0.17 ± 0.03 cd | 92.57 ± 4.69 ab |
| | | DS | 44.70 ± 2.52 abcd | 23.25 ± 1.47 cd | 10.48 ± 1.99 d | 0.15 ± 0.03 d | 78.43 ± 3.95 cd |
| IMC67 | D19 | WW | 46.63 ± 4.53 abc | 26.05 ± 2.34 bc | 17.51 ± 1.18 a | 0.24 ± 0.02 ab | 90.19 ± 4.62 ab |
| | | DS | 41.13 ± 4.45 cd | 21.58 ± 1.94 d | 15.02 ± 1.22 abc | 0.24 ± 0.02 ab | 77.74 ± 4.33 cd |
| | D27 | WW | 45.17 ± 3.06 abc | 28.94 ± 2.22 ab | 16.30 ± 1.74 ab | 0.22 ± 0.03 abc | 90.41 ± 4.54 ab |
| | | DS | 38.50 ± 2.26 d | 23.36 ± 1.66 cd | 13.82 ± 1.58 bcd | 0.22 ± 0.02 ab | 75.67 ± 4.01 cd |

WW, well-watered; DS, water-deficit stress; D19, 19-day water-deficit stress; D27, 27-day water-deficit stress. Average values (*n* = 6). Significant differences (*p* < 0.05) using two-way ANOVA (progeny × stress). Different letters indicate significant differences according to Tukey's HSD test.

It was found that the stem was the organ most affected by water deficit, with IMC 67 OP half-sib progenies showing a lower biomass accumulation starting from day 19 of stress (D19). In contrast, IMC67 × PA121 and SCA6 × PA 121 full-sib families reduced biomass production only when plants were subjected to 27 days of stress (D27). Furthermore, only IMC67 OP half-sib progenies exhibited lower leaf biomass after 27 days (D27) without watering. No significant differences were found in the root/shoot ratio among the evaluated treatments and family progenies. However, it was observed that in the stress treatment (DS), SCA6 × PA121 exhibited higher biomass by 5.43% in plants that went without irrigation for 19 days (D19), while in IMC67 × PA121 full-sib family and IMC67 OP half-sib progenies, it remained unchanged and was lower than well-watered plants (WW), respectively.

### 3.4. Plant Cd Content

It was observed that the highest concentrations of Cd were found in the stem and leaves of the cacao plants, while the roots had the lowest concentration (Figure 2). Regarding stress responses, plants of the IMC67 × PA121 full-sib family progeny showed a significant increase in Cd concentration in the leaves (30.80%) (Figure 2a) and stem (13.89%) (Figure 2b) when subjected to 19 (D19) and 27 (D27) days of stress, respectively. On the other hand, plants of the IMC67 OP half-sib progenies exhibited a reduction of 34.09% and 26.91% in

Cd concentration in the roots during days 19 (D19) and 27 (D27) of water-deficit stress, respectively (Figure 2c).

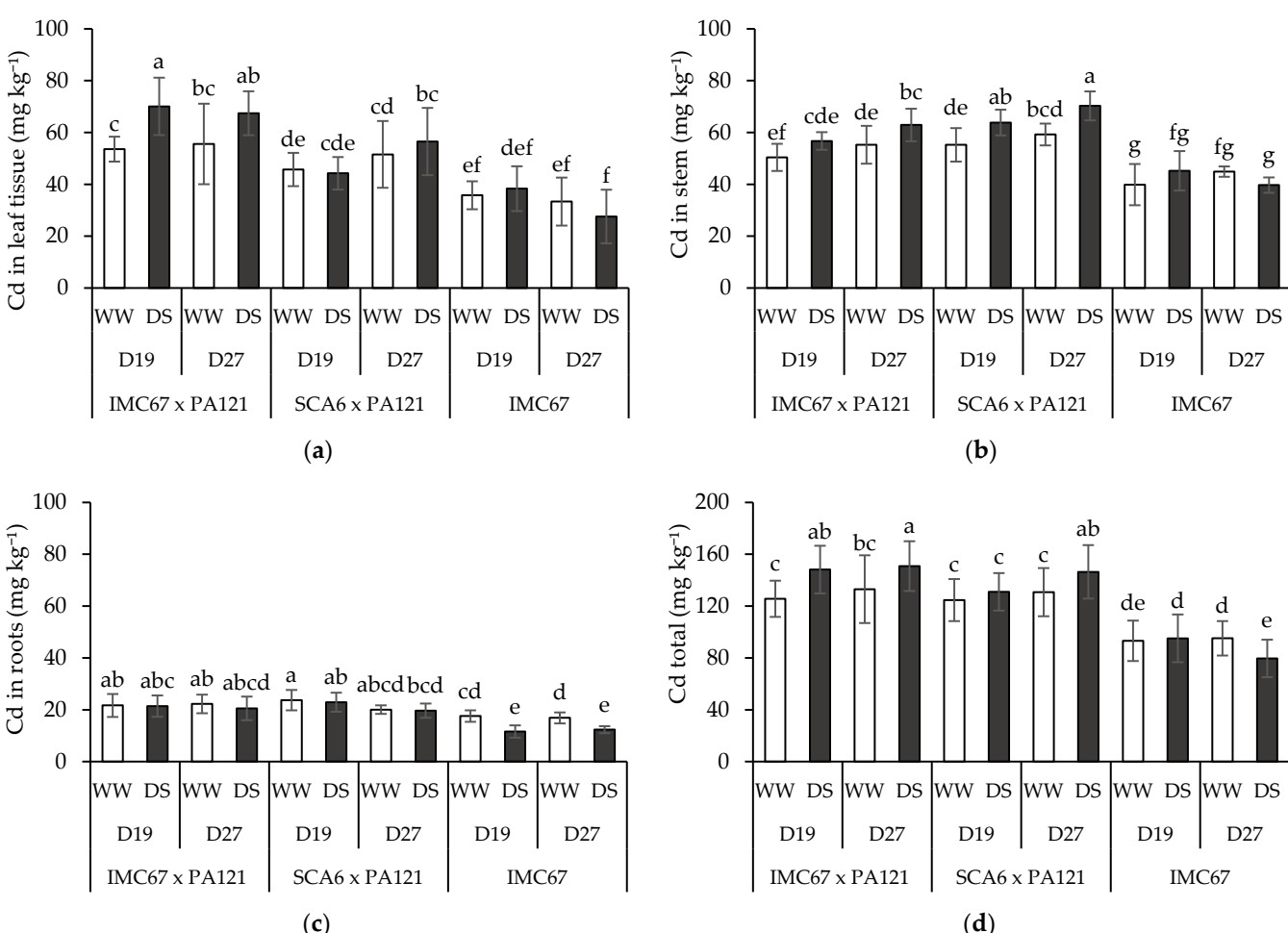

**Figure 2.** Cd concentration in (**a**) leaves, (**b**) stem, (**c**) root, and (**d**) total of three family cacao progenies under water-deficit stress. WW, Well-watered; DS, water-deficit stress; D19, 19-day water-deficit stress; D27, 27-day water-deficit stress. Average values ($n$ = 3). Significant differences ($p < 0.05$) determined by two-way ANOVA (progeny × stress). Different letters indicate significant differences according to Tukey's test.

In the case of the SCA6 × PA121, a significant increase in Cd concentration was observed only in the stem when subjected to 19 (15.57%) and 27 (18.70%) days of water-deficit stress (Figure 2b). These differences in Cd accumulation in the different organs of the cacao plants from each progeny had a significant impact on the total Cd concentration in the stress treatments. Specifically, an increase in total Cd concentration was observed in the IMC67 × PA121 full-sib family progeny at D19 and D27. An increase in Cd content was also observed in SCA6 × PA121 progeny at D27, while a reduction in total Cd concentration was seen in the IMC67 OP half-sib progenies at D27 (Figure 2d).

Leaves accumulated the highest amount of Cd in the plant, followed by the stem and roots. No significant differences were found in Cd accumulation in the roots between IMC 67 × PA121 and SCA 6 × PA 121 full-sib families when subjected to water-stress treatments. However, in plants of the IMC 67 OP half-sib progenies, a significant reduction in Cd accumulation in the roots was observed when subjected to water-deficit stress for 19 (D19) and 27 (D27) days (Figure 3c). Additionally, a 30.77% decrease in Cd accumulation in the stem of IMC 67 OP half-sib progenies under stress for 27 days (D27) was found compared

to well-watered plants (WW) (Figure 3b). No changes in Cd accumulation in the stem were observed for other genotypes and stress levels.

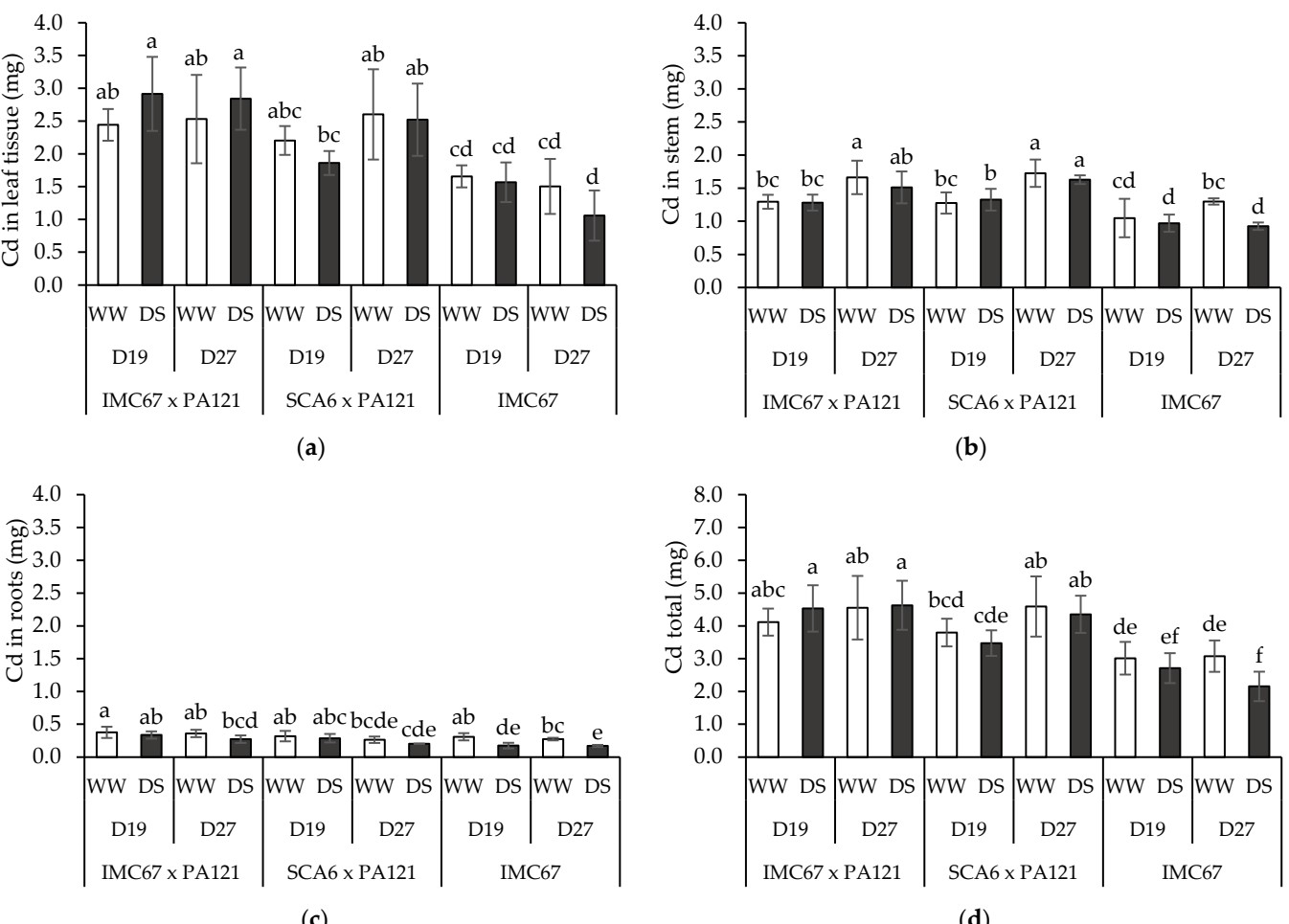

**Figure 3.** Cd accumulation in (**a**) leaves, (**b**) stem, (**c**) roots, and (**d**) total of three family cacao progenies under water-deficit stress. WW, Well-watered; DS, water-deficit stress; D19, 19-day water-deficit stress; D27, 27-day water-deficit stress. Average values (*n* = 3). Significant differences (*p* < 0.05) determined by two-way ANOVA (progeny × stress). Different letters indicate significant differences according to Tukey's test.

Regarding the leaves, no significant differences were found in Cd accumulation between treatments on both day 19 and day 27 of stress. However, there was a trend towards reduced Cd accumulation in the leaves of the IMC 67 OP half-sib progenies when subjected to 27 days of stress, with a decrease of 26.77% (Figure 3a). Overall, the reduction in Cd accumulation in all three organs of IMC 67 plants contributed to a 29.03% decrease in total Cd accumulation in the plant (Figure 3d). No significant differences were found in total Cd accumulation between the full-sib families.

### 3.5. Translocation Factor (TF)

The translocation factor (TF) increased when plants of the IMC67 × PA121 full-sib family and IMC 67 OP half-sib progenies were subjected to 27 (D27) and 19 (D19) days of water stress, respectively (Figure 4). Plants of the SCA6 × PA121 did not show significant differences in Cd accumulation between well-watered (WW) and stressed (DS) plants.

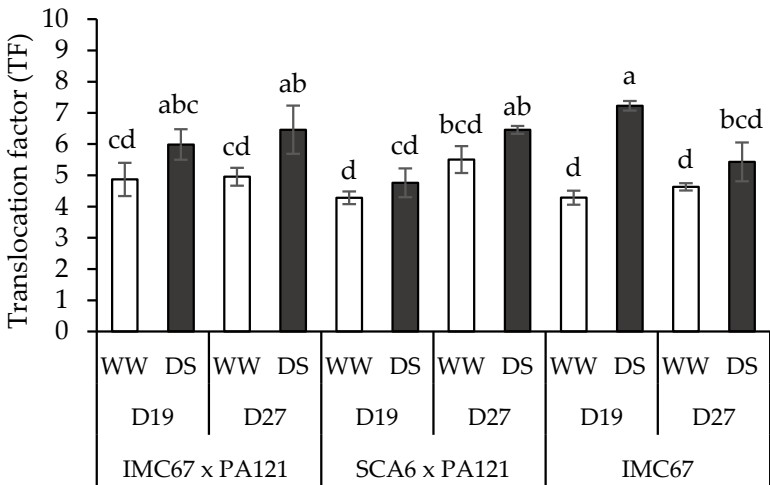

**Figure 4.** Cd Translocation Factor (TF) in three family cacao progenies under water stress. WW, Well-watered; DS, water-deficit stress; D19, 19-day water-deficit stress; D27, 27-day water-deficit stress. Average values (*n* = 3). Significant differences (*p* < 0.05) determined by two-way ANOVA (progeny × stress). Different letters indicate significant differences according to Tukey's test.

## 4. Discussion

The leaf water potential ($\Psi_{leaf}$) serves as an indicator of the hydration status of plants and their capacity to mitigate water loss through transpiration [36]. Several studies suggest that young cacao plants experience moderate stress when $\Psi_{leaf}$ ranges from −0.8 to −1.5 MPa and severe stress below −1.76 MPa [32,37,38].

In accordance with our results, after subjecting the plants to water deficit for 19 days (D19), plants of the SCA6 × PA121 exhibited moderate stress (−1.5 MPa), whereas IMC67 × PA121 full-sib and IMC67 OP half-sib families experienced intermediate stress (−1.9 and −1.8 MPa, respectively), lying between moderate and severe stress (Table 1). After 27 days of water deficit (D27), all three progenies displayed $\Psi_{leaf}$ values indicative of severe stress. In cacao, drought-tolerant progenies exhibit less negative $\Psi_{leaf}$ values than susceptible ones [32,39], implying that the SCA6 × PA121 is more drought-tolerant than IMC67 × PA121 full-sib and IMC67 OP half-sib families.

Stomatal regulation sensitivity to water stress is a crucial physiological trait, determining water loss during transpiration [36]. According to Carr and Lockwoods (2011), *Theobroma cacao* begins partial stomatal closure at a $\Psi_{leaf}$ value of around −1.5 MPa [40]. At this $\Psi_{leaf}$ value (−1.5 MPa), the SCA6 × PA121 family exhibited a 51.43% reduction in stomatal conductance (*gs*) after 19 days of water stress (D19) (Table 2). These findings indicate that stomatal closure likely commenced when leaf water potential was greater than −1.5 MPa, illustrating high sensitivity in stomatal regulation, a significant physiological trait for drought tolerance [36,41].

The correlation between $\Psi_{leaf}$ and photosynthesis (*A*) (r > 0.94) (Figure 5) and the average reduction in stomatal conductance (*gs*) among the evaluated progenies (48.36% and 67.35% at D19 and D27, respectively) suggest a responsive stomatal behavior in regulating water loss through transpiration, consequently reducing *A* due to decreased $CO_2$ consumption. This behavior supports the hypothesis that the photosynthetic response to water deficit is strongly governed by stomatal conductance in cacao plants [30,32] and other perennial species [42–44].

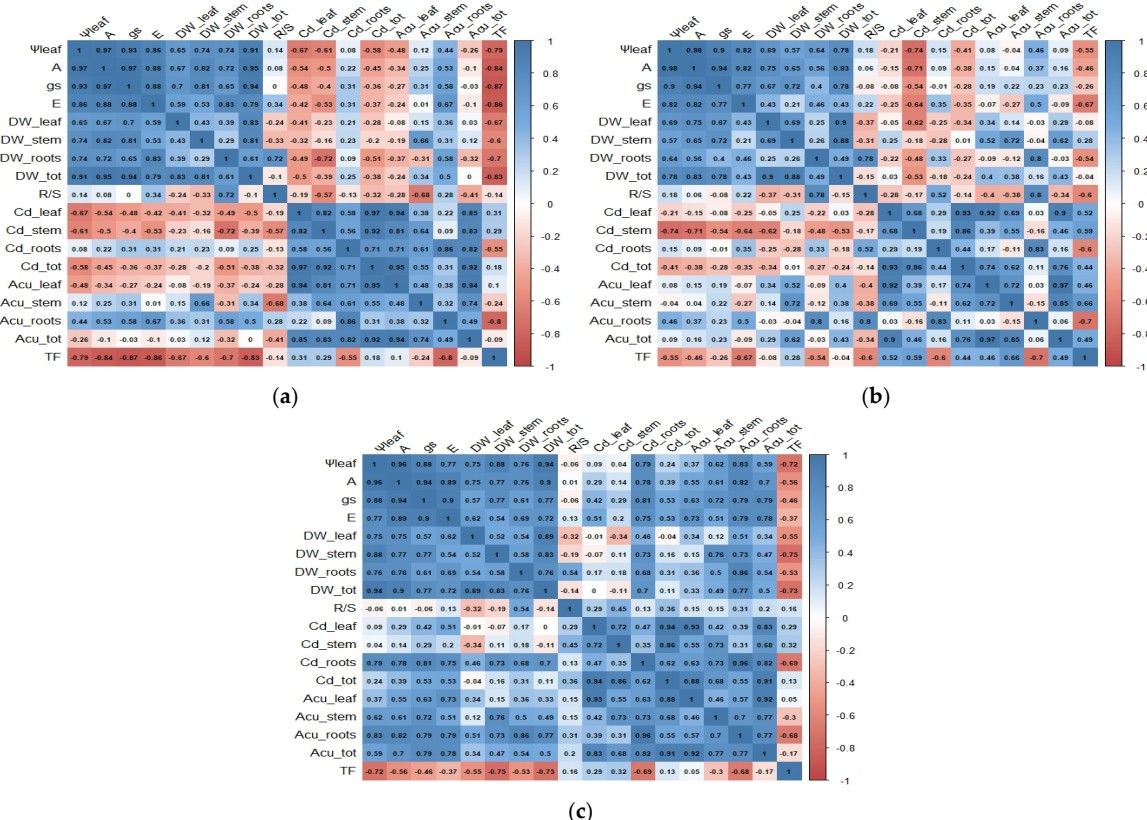

**Figure 5.** Pearson Correlation of physiological, biochemical, and Cd content variables of three family cacao progenies: (**a**) IMC67 × PA121, (**b**) SCA6 × PA121 and (**c**) IMC67. Ψleaf, Leaf water potential; *A*, photosynthetic rate; *gs*, stomatal conductance; *E*, transpiration; DW_leaf, leaf dry weight; DW_stem, stem dry weight; DW_roots, root dry weight; DW_tot, total plant dry weight; R/S, root/shoot ratio; Cd_leaf, Cd concentration in leaves; Cd_stem, Cd concentration in stem; Cd_roots, Cd concentration in roots; Cd_tot, total Cd concentration in plant; Acu_leaf, Cd accumulation in leaves; Acu_stem, Cd accumulation in stem; Acu_roots, Cd accumulation in roots; Acu_tot, Cd accumulation in plant; TF, Cd translocation factor.

Cadmium (Cd) exists in soil primarily as $Cd^{+2}$ ions or in various organic and inorganic complexes [26,45,46]. Soil is considered contaminated by Cd when total content exceeds 3 mg $kg^{-1}$ [47,48] or when the bioavailable Cd concentration surpasses 0.001 mg $kg^{-1}$ [33]. According to Bravo et al. (2021), the maximum permissible concentration of soluble Cd in soils for cacao cultivation is 0.08 mg $kg^{-1}$ [24]. In the soil used for this experiment, a soluble Cd concentration of 0.356 mg $kg^{-1}$ was determined, indicating that cacao plants grew in soil with Cd contamination.

Cd absorption and accumulation in cacao plants are influenced by various factors, including soil pH, Cd availability in the soil, genotype, and plant age [49–51]. In this experiment, it is assumed that there were differences between the treatments in two of the most significant factors affecting Cd absorption in cacao plants: genotype and Cd availability in the soil [50]. The latter variable could have decreased due to the reduced water availability in the soil under water-stress treatments [52,53].

Plants of the IMC67 × PA121 full-sib family displayed the highest Cd accumulation in the plant, averaging 4.35 mg in well-watered plants (WW), followed by SCA6 × PA121 with an average of 4.20 mg, and IMC67 OP half-sib with 3.05 mg, showing significant differences (Figure 3). These results align with Lewis et al. (2018), who classified IMC67 as a low Cd accumulator genotype [10]. The authors suggest that the low Cd accumulation in the IMC67 is due to multiple regulation points during Cd transport from the soil to cacao

roots, shoots, and seeds (cotyledons and testa), possibly mediated by a range of ligands and transport proteins [10].

Genetic influence on Cd absorption and accumulation in plants has been reported in crops such as cacao [10,13,54], soybean, and rice [55]. These variations in Cd accumulation among genotypes are attributed to diverse factors, including varietal vigor, root size and morphology, the activity of ion metal uptake transporters on the root surface, the roots' ability to produce exudates binding to metal ions, and the roots' interaction with soil microorganisms [10,56,57]. This study lacks detailed insight into the mechanisms underlying differential Cd accumulation in the evaluated progenies. Nonetheless, the divergent response of progenies can serve as a tool for investigating Cd accumulation mechanisms in future studies.

Cd content, in terms of concentration, tends to increase in plants experiencing water-deficit conditions due to reduced biomass accumulation [22,23]. While no significant correlation was found between Cd concentration and accumulated biomass in different organs (Figure 2), an increase in total Cd concentration was observed in the IMC67 × PA121 full-sib family. This could be explained by a significant decrease in plant dry mass on days 19 and 27 of water stress (Table 3). Similarly, the elevated Cd concentration in SCA6 × PA121 after 27 days of stress could result from reduced biomass accumulation under that stress level.

Despite the lack of correlation between Cd concentration and cacao plant dry mass, significant correlations were observed in the IMC67 × PA121 full-sib family between leaf water potential ($\Psi_{leaf}$) and Cd concentration in leaves (r = −0.67), stems (r = −0.61), and total concentration (r = −0.58) (Figure 5a). Furthermore, a significant correlation was found between $\Psi_{leaf}$ and Cd concentration in stems (r = −0.74) in the SCA6 × PA121 family. These correlations indicate that the plant's water status influences Cd accumulation in its organs, and this relationship is not solely determined by reduced biomass accumulation under water-deficit conditions. This is confirmed in the case of the IMC67 OP half-sib family, in which despite reduced accumulated biomass due to water-stress treatment (Table 3), no increase in plant Cd concentration was observed (Figure 2).

Several factors can diminish Cd absorption and accumulation in plants under water-deficit conditions [17], including root morphology and reduced transpiration due to stomatal closure [53,58,59]. However, this behavior was only observed in the IMC67 OP half-sib, with reduced Cd accumulation in different plant organs (Figure 3).

Importantly, a positive correlation between transpiration (*E*) and total Cd accumulation (r = 0.78) was found in the IMC67 OP half-sib (Figure 5c). This suggests that the significant 64.90% reduction in transpiration in the IMC67 OP half-sib between days 19 and 27 of stress (Table 2) directly impacted Cd absorption and accumulation by cacao plants. This reduced transpiration explains the 21.06% decrease in Cd concentration in roots between day 19 and day 27 of stress (Figure 2). These findings align with previous studies, highlighting the influence of genotype and stress duration on heavy metal accumulation in plants [17,19,22,23]. In other progenies, no significant effect of water deficit on Cd accumulation was evident, possibly due to their transpiration being less sensitive to reduced leaf water potential between days 19 and 27 of water stress (Table 2).

Previous research has demonstrated that water stress can influence both Cd absorption and translocation in plants, as observed in species such as Indian mustard (*Brassica juncea*) and castor bean (*Ricinus communis*) [16]. Additionally, various authors assert that the amount of Cd accumulated and translocated varies by species and genotype [10,13,60,61]. Consistent with this, significant differences were observed in this study between stress levels and progenies in Cd translocation within cacao plants (Figure 4).

Considering that the translocation factor (TF) indicates the ability of selected cacao progenies to transport Cd from roots to the aboveground parts [11,62], it can be affirmed that the evaluated cacao progenies, particularly IMC67 OP half-sib and IMC67 × PA121 full-sib progenies, exhibited greater Cd translocation from roots to aboveground parts when subjected to stress (DS) for 19 and 27 days, respectively, compared to well-watered plants

(WW) (Figure 5). In contrast, the SCA6 × PA121 family showed no significant changes in translocation factor in response to water deficit.

It is pertinent to emphasize that despite plants from progeny IMC67 OP half-sib accumulating the lowest amount of Cd under moderate water-stress conditions ($\Psi_{leaf}$ = −1.8 MPa) (Figure 3d), they exhibited the highest capacity to translocate Cd from roots to aboveground parts (Figure 5) when subjected to 19 days of stress (D19). This is attributed to IMC67 OP half-sib plants maintaining the highest transpiration at this stage of stress, with 12.94% more than the IMC67 × PA121 full-sib and SCA6 × PA121 progenies. Conversely, progeny SCA6 × PA121, which displayed high Cd accumulation, did not exhibit increased translocation under water-deficit conditions. While both IMC67 × PA121 full-sib and SCA6 × PA121 progenies displayed a correlation between the translocation factor and transpiration (Figure 5a,b), it can be observed that for these two progenies, a reduction in transpiration ranging from 50 to 52.94% under water-deficit conditions (Table 2) was insufficient to significantly enhance Cd translocation to aboveground parts (Figure 4). These outcomes suggest that both genotype and plant water status, as well as their interaction, influence Cd translocation capacity.

Typically, Cd content is higher in roots compared to aboveground parts due to roots acting as an effective barrier to limit Cd translocation in plants [11]. For instance, a study by Castro et al. (2015) on 2-month-old cacao plants subjected to different $CdCl_2$ solutions through seed imbibition found greater Cd accumulation in roots than in stems and leaves [9]. However, the presence of Cd accumulation in the aboveground parts of cacao plants indicates that the metal was not entirely immobilized by roots. Additionally, Pereira de Araújo et al. (2017) conducted a study on 4-month-old cacao plants exposed to different Cd concentrations in soil and found that Cd was equally absorbed and accumulated in all plant organs, implying Cd translocation from roots to aboveground parts [8].

In contrast, a study by Fernández-Paz et al. (2021) on 8-month-old cacao root-stocks revealed higher Cd concentrations in leaves (85 mg kg$^{-1}$) compared to roots (40 mg kg$^{-1}$) [12]. While tolerance to heavy-metal-contaminated soils is generally associated with metal retention in root cells to prevent its translocation to aboveground parts [11,60], there are hyperaccumulating plants that transport most of the heavy metal to aboveground parts through the xylem [63]. Although it may seem that this is the case in cacao, greater Cd accumulation in leaves rather than roots is influenced by genotype and plant age [51]. These factors underscore the complexity of Cd accumulation in cacao and emphasize the importance of considering multiple variables in interpreting Cd distribution patterns across different plant tissues.

Cd concentrations found in the leaves of the evaluated progenies are below levels reported in cacao plants under one year of age. According to Fernández-Paz et al. (2021), leaf Cd values ranged from 80 to 100 mg kg$^{-1}$ [12], De Almeida et al. (2022) recorded an average of 200 mg kg$^{-1}$ [11], and Pereira de Araújo et al. (2017) detected concentrations of 388.970 mg kg$^{-1}$ [8]. These differences are attributed to genotype, initial Cd concentrations present in the soil, and the use of highly soluble Cd sources, such as Cd $(NO_3)_2$ and $CdCl_2$, in the mentioned studies. Importantly, no external Cd sources were added to the soil in this experiment, which could influence Cd availability for plant uptake [46].

In the case of cacao plants older than ten years growing in Cd-contaminated soils without external soluble sources, as in this study, Cd accumulations in leaves are considerably lower than concentrations reported in young plants. Previous studies documented maximum values of 5.42 mg kg$^{-1}$ in leaves [10,13,64]. It has been demonstrated in cacao and various species that plant age plays a pivotal role in Cd absorption and accumulation [49,57]. Argüello et al. (2019) found that young cacao plants absorb more Cd than older ones [49]. This has been associated with deeper roots containing a lower amount of bioavailable Cd, higher calcium (Ca) content in mature plantations potentially blocking Cd absorption [49], greater growth in young plants, and higher biomass in older trees, leading to reduced Cd concentration in plant tissues [65].

On the other hand, young plants exhibit less developed and less lignified root systems compared to older plants. Consequently, apoplastic barriers, such as Casparian bands are not fully suberized in young plants, particularly in young roots or apical root areas [66]. According to Song et al. (2019), in *Cunninghamia lanceolata*, Casparian bands fulfill their barrier function during the secondary stage of endodermis development, i.e., after lignin and suberin are deposited in the cell wall [67]. Thus, it is plausible that young plants have a reduced capacity to discriminate against Cd in roots before it enters the xylem and is transported to leaves [66].

According to Mingorance et al. (2007), a translocation factor (TF) > 1 indicates that cacao plant organs are enriched with Cd [68]. In the case of the evaluated progenies in our study, a TF above 4 was observed, signifying that the aboveground organs of these young cacao plants were significantly enriched with Cd. Nevertheless, it is important to note that the transport of Cd from the roots to the aerial part may exhibit variations after grafting a scion onto the rootstock, influenced by the relationship between the two parts [12].

Cd translocation in plants is associated with metal retention in roots and loading activity in the xylem [69,70]. Based on the observed translocation factor (TF) values in the progenies of this study, it can be inferred that there is no efficient restriction in Cd movement to the vascular cylinder, suggesting that roots might not be fully suberized due to endodermis immaturity [71].

In young roots or apical root areas where apoplastic barriers, such as Casparian bands, are not fully developed, a substantial amount of absorbed Cd may be directly transferred from the xylem to the phloem in woody plant parts, such as the stem [72]. A study conducted by Engbersen et al. (2019) on different cacao genotypes found that a substantial portion of the Cd entering the roots could be transported from stems and branches to the cacao seed without passing through the leaves [73]. This demonstrates that the stem can be an organ in which the cacao plant accumulates this metal. Findings by Moore et al. (2020) suggest that Cd in the stem may be sequestered through biochemical pathways similar to those occurring in roots [74].

Once Cd reaches the vascular cylinder, this metal can re-enter the stem's apoplast before being loaded into the xylem vessels [75]. Studies have detected the presence of Cd in the walls of xylem vessels in various species [76], which could explain the high Cd accumulation in the stems of plants belonging to the IMC67 × PA121 full-sib and SCA6 × PA121 progenies. Once in the xylem, this Cd can bind to ligands like phytochelatins and carboxylic acids in the vacuole and cell walls [60,77], and its loading capacity depends on transporter activity [60].

Furthermore, research in other species, such as red maple, has demonstrated that the combined stress from drought and heavy metals can reduce the density of xylem vessels, resulting in an increase in parenchyma cells in the xylem [78]. This increase in parenchyma under combined stress conditions could explain the storage of Cd translocated from the root to the stem in the IMC67 × PA121 full-sib and SCA6 × PA121 progenies. In the case of the IMC67 × PA121 full-sib, this is especially pertinent since during moderate stress (D19), Cd was primarily concentrated in the leaves, whereas under severe stress (D27), Cd concentration shifted to the stem (Figure 2), possibly due to a reduction in vessel density and an increase in xylem parenchyma cells [78].

These findings underscore the complexity of Cd uptake and translocation processes in plants as well as the influence of genetic and environmental factors, like water deficit, on these mechanisms.

## 5. Conclusions

Water deficit can lead to an increase in Cd concentration in cacao plants due to reduced biomass accumulation under stress. However, Cd absorption and accumulation in the plant can also be influenced by limited transpiration, depending on the genotype and stress level.

The accumulation of Cd is largely determined by genotype, and the IMC67 OP half-sib progeny stands out as a low-accumulation genotype. However, it is important to note that

under conditions of moderate water deficit, an increase in Cd translocation from the roots to the aerial part of the plant was observed.

It is relevant to highlight that the movement of Cd from the roots to the aerial part may vary in an ungrafted plant compared to a grafting system between scion and rootstock. Therefore, a study is required to assess this translocation in grafts with these specific progenies to determine if it occurs similarly. Progeny SCA6 × PA121 exhibits promising traits as a potential rootstock due to its ability to withstand water stress and maintain stable Cd accumulation. However, further evaluation is needed regarding Cd accumulation and its translocation from the rootstock to the grafted scion under water-deficit conditions. Additionally, it is important to take into account other factors, such as the age of the plant, which may influence the amount of Cd absorbed and transported from the rootstock to the canopy.

This study initiates the discussion of important considerations for the proper selection of cacao genetic materials in regions characterized by droughts and high soil Cd concentrations. A deeper understanding of the interaction between water stress and Cd accumulation in cacao is warranted, potentially opening new avenues for enhancing the adaptation and performance of this crop under challenging conditions.

For future research aimed at assessing the interaction between water deficit and Cd accumulation in cacao, it is recommended to incorporate additional molecular and biochemical analyses. These analyses could encompass the investigation of the behavior of antioxidant enzymes, the production of phytochelatins, metallothioneins, and carboxylic acids. This approach would lead to a more comprehensive understanding of the mechanisms involved in the transport and translocation of Cd in cacao plants.

**Author Contributions:** Conceptualization, A.O.-Á., S.M., E.A.S.-A., C.R.-M., X.A. and Á.M.C.-M.; methodology, A.O.-Á., S.M., E.A.S.-A., C.R.-M., X.A. and Á.M.C.-M.; software, A.O.-Á. and E.A.S.-A.; validation, A.O.-Á., E.A.S.-A. and Á.M.C.-M.; formal analysis, A.O.-Á. and E.A.S.-A.; investigation, A.O.-Á., S.M., E.A.S.-A., C.R.-M., X.A. and Á.M.C.-M.; resources, C.R.-M. and X.A.; data curation, A.O.-Á. and E.A.S.-A.; writing—original draft preparation, A.O.-Á.; writing—review and editing, A.O.-Á., S.M., E.A.S.-A., C.R.-M., X.A. and Á.M.C.-M.; supervision, Á.M.C.-M.; project administration, C.R.-M. and Á.M.C.-M.; funding acquisition, C.R.-M. and X.A. All authors have read and agreed to the published version of the manuscript.

**Funding:** This research was funded by the European Union via the European Commission under the DeSIRA initiative as part of the Clima-LoCa project, Grant Contract FOOD/2019/407-158. Its contents are the sole responsibility of the authors and do not necessarily reflect the views of the European Union.

**Data Availability Statement:** The data presented in this study are available on request from the corresponding author. The data are not publicly available because the plant materials we use are currently undergoing evaluation as part of a cocoa genetic improvement program.

**Acknowledgments:** Acknowledgment goes to researchers Isidro Beltrán and Eliseo Polanco from the Nataima Research Center of AGROSAVIA, who provided the materials and field personnel for the setup of plants and the development of field trials. Additionally, we would like to thank the National Germplasm Banks (SBGNAA) for supplying plant material. We extend our thanks to Camila Hernández for obtaining the cacao progenies through controlled crosses between cacao accessions from the SBGNAA.

**Conflicts of Interest:** The authors declare no conflict of interest.

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
