# Peer review of "Cadmium Accumulation in Cacao Plants (Theobroma cacao L.) under Drought Stress"

_agronomy, doi:10.3390/agronomy13102490_

Round 1

Reviewer 1 Report

The paper determined Cd accumulation under water deficit conditions in cocoa rootstock candidates, since the Cocoa (Theobroma cacao L.) is one of the world’s most important crops and has been identified as a significant dietary source of Cd. This is a very important worldwide issue and thus research such as this one is more than valuable. Cd accumulation mitigation in the edible crops is still not well understood so presented research tackles an urging question. This research reveals some fine differences influenced by rootstock genotypes, which shift rootstock selection towards the drought-resistant and low Cd-accumulating ones.

Line 13: Please finish this sentence, as the abstract is supposed to be clear on a stand-alone basis. 

Lines 15-17 are not comprehensible, please rewrite it. What do you mean by Materials with potential?

Line 85:  this chapter.. do you mean this paper?

Line 226: Factor de translocación de Cd (TF), why not in English? 

Tables 1 and 2: uniform font and size, some are capitalized others are not.

Also, please uniform the referencing. Add the number instead of the year for Castro et al. (2015) and similar. 

The discussion section is extremely long, please consider shortening it significantly. 

Minor revision is required.

Reviewer 2 Report

The MS provides a comprehensive overview of Cd accumulation under water deficit conditions. It cites previous research to support its claims and findings and effectively emphasizes for plants facing cadmium-induced stress. Generally, it is a good paper and deserves to be published. 

Some minor mistakes of spelling should be carefully check and correct

Reviewer 3 Report

The text require major revision, with at least discussion of Cd accumulation in the fruit.

Some minor comments>

Abstract.

Need to be re-adjusted. Which biological question did you ask? What is the scoentific value of your results? It is not necessray to decribed small details like name of greenhouse etc. But provide reader general information about results.

Ψleaf?? Not all reader know this abbreviation.

Please, provide only general summary, avoid details.

Line 31: “Cadmium (Cd) accumulation, a highly toxic heavy metal, in agricultural crops…” Cadmium or accumulation is highly toxic?

Line 35: citation N2 is not about dietary. Please, carefully read original paper.

Lines 85-89: please, decribe “biological aim” of your study: to find way to reduced Cd contents in the end products? Or others?

Line 103: AGROSLAVIA 2021? Citation?

Lines 107 – 109: why plants have a stavation period? One month is 30/31 days, not 60. Please, clarify setup.

Lines 117- 122: is this amount per 40 kg? Or per 1 kg of the soil?

Line 119: “Transfer ionic” – please, provode the link. Which pH, How did you prepare? Have you consider that ascorbic acid at pH 5.3 rapidly convert to lactone?

Line 124: why cc, not ml?

Figure 1: rather results. Why there are such variation in all parameters at day 8, 17 and 27?

What were the real temperatures in greenhouse? Did temeprature in weather station mean “official” or real?

Lines 216- 225: it will be nice to measure Cd contents in the real organs as fuit, what have been used in industry. Please, explain why leaves and root are so important.

In the text I found figure 1, and thereafter (245) figure 3. Maybe you mean Table 1?

It will be more logical to move part 3.4 instead 3.1?

Table 3: what is dry matter accumulation? Per what? Plant? How it is linked with fresh weight? Please, provide all information.

Figure 2: fresh or dry weight?

Line 396: soil solution??

Line 573: conclusión can not conatins “can”, it is based on results!  

Line 581: “Progeny 1233” ¿ genotype 1233?

moderate

Reviewer 4 Report

In my opinion, this is an interesting manuscript, considering that two major abiotic stresses are studied in a same manuscript. Indeed, at the field level, different abiotic factors may happen in parallel.

I provide the following recommendations to improve the manuscript:

1) When explaining about rootstocks, the authors cited several articles that are not current.

Thus, recent articles on use of grafting and rootstocks for cadmium-exposed plants should be cited in order to provide a current picture to the readers.

2) In the abstract, the authors mention "Materials with potential". This “potential” should be better explained.

3) Molecular and biochemical analyses related to abiotic stress tolerance and response (as well as molecular mechanisms of cadmium accumulation and transport) are very important but they have not been reported in this manuscript. These include antioxidants, phytochelatins, omic approaches, etc. Thus I recommend the authors at least dedicate some sentences on the Discussion or Conclusion to suggest those analyses for further studies.

4) Please be careful regarding the writing. Some sentences are written in other language (for example: see the title of the topics 2.11 and 3.5 - these are not in English). 

4.1) Has this manuscript been published elsewhere as a chapter thesis, book chapter book or something like that? I ask this because the authors wrote "The objective of this chapter was..." (line 85). Please be careful to avoid plagiarism if the current content has been used in other documents.

5) In the last paragraph of the Methods topic 2.11, the authors explain why they determined TF index. Honestly I don't think this is relevant within a Methods section context. You might place such paragraph in another manuscrip section instead.

6) In 2.11, is the abbreviation TF (as you mentioned in the main text) or FT (like you mentioned in the equation)? Please fix it.

7)  The word "cadmium" should be abbreviated in different sentences. For example, in the lines 239 and 241, "cadmium" is not abbreviated.

8) Please revise the word "intolerant" (line 51). It sounds uncommon to me within this research topic. Do you mean sensitive genotypes?

Round 2

Reviewer 3 Report

few very minor corrections

Authorsb carefully answered all questions and I think current text can be accepted.